# Tracking Brownian motion in three dimensions and characterization of individual nanoparticles using a fiber-based high-finesse microcavity

Larissa Kohler[1✉], Matthias Mader [2,3], Christian Kern[4,5], Martin Wegener [4,5] & David Hunger [1,6✉]

The dynamics of nanosystems in solution contain a wealth of information with relevance for diverse fields ranging from materials science to biology and biomedical applications. When nanosystems are marked with fluorophores or strong scatterers, it is possible to track their position and reveal internal motion with high spatial and temporal resolution. However, markers can be toxic, expensive, or change the object's intrinsic properties. Here, we simultaneously measure dispersive frequency shifts of three transverse modes of a high-finesse microcavity to obtain the three-dimensional path of unlabeled $SiO_2$ nanospheres with $300\,\mu s$ temporal and down to $8\,nm$ spatial resolution. This allows us to quantitatively determine properties such as the polarizability, hydrodynamic radius, and effective refractive index. The fiber-based cavity is integrated in a direct-laser-written microfluidic device that enables the precise control of the fluid with ultra-small sample volumes. Our approach enables quantitative nanomaterial characterization and the analysis of biomolecular motion at high bandwidth.

[1] Karlsruher Institut für Technologie, Physikalisches Institut, Wolfgang-Gaede-Str. 1, 76131 Karlsruhe, Germany. [2] Fakultät für Physik, Ludwig-Maximilians-Universität, Schellingstraße 4, 80799 München, Germany. [3] Max-Planck-Institut für Quantenoptik, Hans-Kopfermann-Str. 1, 85748 Garching, Germany. [4] Karlsruher Institut für Technologie, Institut für Angewandte Physik, Wolfgang-Gaede-Str. 1, 76131 Karlsruhe, Germany. [5] Karlsruher Institut für Technologie, Institut für Nanotechnologie, Hermann-von-Helmholtz-Platz 1, 76344 Eggenstein-Leopoldshafen, Germany. [6] Karlsruher Institut für Technologie, Institut für QuantenMaterialien und Technologien, Hermann-von-Helmholtz-Platz 1, 76344 Eggenstein-Leopoldshafen, Germany. ✉email: larissa.kohler@kit.edu; david.hunger@kit.edu

The development of optical sensors for the detection and analysis of individual nanosystems and their motional dynamics is of great importance. Ideally, a sensor would give direct access to the intrinsic optical material properties such as polarizability and absorptivity, differentiate functional properties such as, e.g., the drug load of nano-carriers or the folding state of proteins, and allow to monitor the Brownian motion to reveal diffusion dynamics of individual nanosystems. With fluorescence microscopes it is possible to obtain spatial information beyond the diffraction limit and to resolve fluorophore locations down to the nanometer scale[1–3], which enables the tracking of individual nanosystems[4–6]. Without labeling, it is possible to track nanosystems by the detection of the Rayleigh-scattered light[7] and using interference techniques to extract the weak signal from the background noise[8,9]. The weak signals enforce the integration over time or the labeling of the nanosystem with a strong scatterer to reach better temporal resolution[10]. Other approaches which enable single nanoparticle detection are evanescent biosensors using, e.g., nanoplasmonics[11,12], tapered fibers[13], or whispering gallery mode resonators such as microspheres[14,15], which enable the real-time detection, e.g., of adsorption events[16–19] and quantitative particle sizing[20]. However, those sensors are limited to the evanescent near-field. To achieve the monitoring of unconstrained diffusive motion, sensing of nanosystems far away from the sensor surface is desired. Open-access microcavities have recently been introduced as a promising alternative platform for refractive index sensing[21] or nanoparticle trapping[22]. The full access to the cavity mode allows for optimal overlap of the sample with the cavity mode and enables quantitative nanoparticle characterization. So far, focused-ion-beam milled structures on planar substrates have been used, with the challenge to provide controlled flow through a specific cavity, the need for free-space optics, and limited cavity finesse. Furthermore, as a general aspect, cavity-enhanced measurements typically rely on probing isolated resonances which do not provide much spatial information about the sample. Exceptions are scanning microcavities, where a cavity mode is spatially raster-scanned across a fixed sample[23,24], but they are not capable to track the fast Brownian motion of nanosystems. Also, cavities with special geometries that lead to degenerate transverse modes[25,26] can have imaging capabilities; however, so far they lack the sensitivity required for nanoscale samples. It is therefore highly desirable to extend the capability of microcavities for ultra-sensitive quantitative characterizations towards the analysis of motional dynamics and position tracking.

Here, we use a high-finesse open-access microcavity to demonstrate quantitative nanoparticle characterization and introduce a novel technique to perform three-dimensional (3D) position tracking of nanoparticles dispersed in water. The device is long-term stable such that we can obtain an extended statistics from several hundred single-particle transits. This allows us to measure the particle's polarizability and the temporal variation of the sample over time. If the particle size is measured in addition, its effective refractive index can be determined. Therefore, we introduce a novel scheme for particle tracking. By simultaneously measuring the frequency shifts of three different transverse modes of the cavity, we are able to infer the instantaneous 3D coordinates of a nanoparticle and resolve its trajectory. From transient tracks we can deduce the particle's diffusivity and thereof its hydrodynamic radius, such that together with the measured polarizability, we can infer the key dispersive properties of individual nanosystems. Particle tracking adds a central functionality to cavity-based sensing, opening up prospects for studies of diffusion dynamics and accurate characterizations of unlabeled nanosystems.

## Results

**Integrated microfluidic cavity setup.** Our Fabry-Pérot cavity system is based on two laser-machined and mirror-coated optical fibers that form a microcavity between the two fiber end facets[27,28], see Fig. 1a. The fibers are inserted into a precision ferrule for lateral alignment and are mounted on shear piezo-electric actuators outside the ferrule (see Supplementary Fig. 1) for fine-tuning of the mirror separation and thereby the cavity resonance condition. The cavity standing wave is located in a microfluidic channel, which is defined by a 3D laser-written polymer structure (see Supplementary Fig. 2) that forms a 200-μm-diameter channel transversing the cavity, thus allowing for a laminar, precisely controllable flow through the microfluidic device, see Fig. 1b. The cavity can be probed by coupling light into the single-mode (SM) fiber, and we detect the transmitted light emerging from the cavity through a multi-mode (MM) fiber. When operated in air, the cavity finesse is as high as $\mathcal{F} = 90,000$, while when immersed in distilled and filtered water, we observe a finesse of up to $\mathcal{F} = 56,700$. The reduced finesse in water is expected and consistent with the change in mirror reflectivity due to the higher refractive index of water compared to air, and the absorption loss in water. At the mirror separation used in the experiments, the quality factor amounts to $Q = 1.0 \times 10^6$, the mode volume is $V_{mode} = 40(\lambda/n_m)^3$, and the mode waist is as small as $w_0 = 1.5$ μm. The figure of merit for the sensitivity of this cavity, $\mathcal{C} \propto Q\lambda^3/V_{mode} \propto \mathcal{F}\lambda^3/(\pi w_0^2)$, is a factor ~100 greater than those of optimized WGM microspheres[16,29] and on par with the best reported microtoroid sensors[19,30–32] in water, with the additional advantage of a fully accessible cavity mode which avoids the permanent modification of the cavity due to binding events, and which permits the observation of free, unconstrained diffusion. Furthermore, the cavity resonance can be tuned to a desired wavelength, e.g., to match that of a fixed probe laser, or

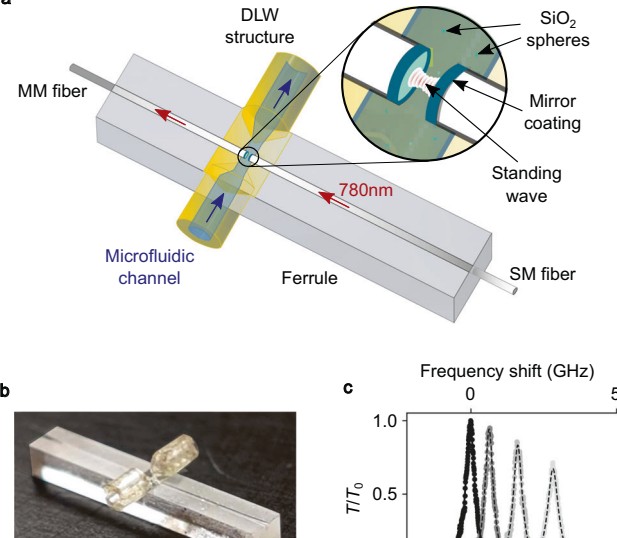

**Fig. 1 Fabry-Pérot microcavity device and measurement signal. a** Schematic setup showing the cavity and its integration into a glass ferrule with a direct-laser-written (DLW) structure (yellow) forming the microfluidic channel. The cavity consists of a single-mode (SM) and a multi-mode (MM) glass fiber with dielectric mirrors attached to their end facets. **b** Photograph of the microfluidic device. **c** Examples of the cavity transmission signal $T/T_0$ for an empty cavity (black) and with nanoparticles present at increasing spatial overlap with the cavity mode (decreasing gray value). An increasing frequency shift, decreasing peak transmission, and increasing linewidth are visible.

can be used for cavity-enhanced spectroscopy. In the experiments described below, we probe the cavity with a grating-stabilized diode laser of wavelength $\lambda = 780$ nm and detect the transmitted light by an avalanche photodetector. The cavity length is modulated at a frequency of 3–10 kHz to repeatedly sample the fundamental cavity resonance. For each occurring resonance, we measure the location and amplitude of the peak to infer the cavity frequency shift and the transmitted power as a function of time (see Supplementary Fig. 4). For all experiments, we flooded the microfluidic cell with the nanoparticles dispersed in double distilled water and then turned the pressure off to stop the flow. After 10 min to 1 h of waiting time we started the measurements. We choose a low particle concentration such that the cavity is empty most of the time, i.e. the time interval between particle transit events is about 10 s and each event lasts on average 0.47 s (see Supplementary Fig. 6). Figure 1c exemplarily shows snapshots of the cavity transmission signal when a single nanoparticle enters the cavity with increasing spatial overlap with the cavity mode. In the following, we present results which were measured by two different cavity setups (1,2) with two different samples (A, B) (see Supplementary Note 1).

**Single nanoparticle sensing and quantitative analysis**. We investigate $SiO_2$ nanospheres with a narrow size distribution (sample A: radius $63 \pm 2$ nm, sample B: $60 \pm 2$ nm) as a model system to characterize and calibrate our device. When a nanosphere with effective refractive index $n_{eff}$, hydrodynamic radius $r_{hydr}$ and volume $V_{NP}$ enters the cavity standing wave light field with normalized intensity distribution $I_{cav}$ and filled with a liquid of refractive index $n_m$, its polarizability $\alpha = 4\pi r_{hydr}^3 \epsilon_0 (n_{eff}^2 - n_m^2)/(n_{eff}^2 + 2n_m^2)$ produces a relative frequency shift $\Delta\nu/\nu = \alpha(\int_{V_{NP}} I_{cav} dV_{NP})/(2\epsilon_0 V_{mode})$ of the cavity resonance. The amount of shift depends on the electric field strength penetrating the nanosphere and is therefore maximal at the antinode and minimal but non-zero at the node since the nanosphere has a non-negligible extension. In addition, the non-absorbing $SiO_2$ nanosphere leads to a decrease of the cavity peak transmission due to additional loss from Rayleigh scattering, $S = |\alpha|^2 k^4/(6\pi\epsilon_0^2)$. Figure 2a shows an exemplary time trace of the cavity resonance shift and the corresponding transmission change produced by a single nanoparticle (sample B, cavity 2, see Supplementary Fig. 5 for details on data evaluation). The fluctuations emerge from the particle's Brownian motion through the standingwave cavity field as schematically depicted in Fig. 2b. We can measure many such events and correlate the frequency shift and the transmission change as depicted in Fig. 2c. Here, the data of 330 transit events are shown (sample A, cavity 1). The transmission change and the frequency shift signals depend on the cavity mode geometry in the same way, such that the slope of the signal is independent of it[20]. This allows us to evaluate the polarizability in a quantitative manner that is immune against systematic errors of the cavity geometry. If furthermore the particle size is known or measured (see Section "particle tracking"), one can infer the refractive index of the material. For this purpose, we compare the observed mean values of the correlation signal with the expected signal for a spherical particle with the hydrodynamic size given by the manufacturer and use the effective refractive index as a free fit parameter (see Supplementary Note 2). From this, we can precisely determine the effective refractive index, and for a nanoparticle hydrodynamic radius of $r_{hydr} = 71.5$ nm we find $n_{eff} = 1.41 \pm 0.01$ (see Supplementary Fig. 7). The measured effective refractive index is smaller than the value of the bare particle ($n_{SiO2} = 1.42$ provided by the manufacturer, sample A) since our measurement is sensitive to the nanoparticle including its hydration shell (see below). The

effective polarizability of sample A amounts to $\alpha_{eff}/(4\pi\epsilon_0) = (14.5 \pm 1.8) \times 10^3$ nm$^3$ being larger than the intrinsic polarizability of $\alpha_{SiO2}/(4\pi\epsilon_0) = (11.2 \pm 3.3) \times 10^3$ nm$^3$ of the bare nanoparticle.

When a single nanoparticle diffuses in the cavity light field, the probability to transit through the electric field maximum and accordingly to produce the maximal frequency shift is small. This can be quantified by the density of states of available positions that produce a certain frequency shift (see Supplementary Note 3). Figure 2d shows histograms of frequency shifts from nanoparticle events, which were measured successively without changing the nanoparticle solution inside the microfluidic cell in between. The histograms can be fitted by the modeled density of states expected for single nanoparticles with the refractive index as the only free parameter (sample B, cavity 2). Already for a single nanoparticle transit (black data), a good agreement is found and the polarizability or the refractive index (if $r_{hydr}$ is known) can be inferred. The statistics improves for intermediate measurement times, and averaging over 59 transits yields the best agreement with $\alpha_{eff}/(4\pi\epsilon_0) = (19.0 \pm 1.0) \times 10^3$ nm$^3$ and $n_{eff} = 1.43 \pm 0.02$ (blue data). This result for the refractive index is consistent with the expected value for the bare particle refractive index of sample B (see Supplementary Fig. 10). More details are given in Supplementary Fig. 8. For measurement times longer than 2 h, we observe an increase in larger frequency shifts, extending to values up to two times larger than the expected maximum single-particle frequency shift (gray data). This observation is consistent with the agglomeration of nanoparticles into dimers and demonstrates the capability to monitor temporal changes of the sample properties in a sensitive manner.

For the calculation of the refractive index from the data shown in Fig. 2d we measured the value for the nanoparticle radius of $r_{hydr} = 75.3 \pm 2$ nm in aqueous solution by dynamic light scattering (DLS), hence representing the hydrodynamic radius, which is 15.3 nm larger than the geometric radius of the solid glass sphere of $r_{SiO2} = 60.0 \pm 2$ nm, which was obtained from transmission electron micrographs (see Supplementary Fig. 10). When assuming that our data originates only from the $SiO_2$ nanosphere with $r_{SiO2} = 60$ nm, the observed frequency shifts would correspond to a refractive index of $n_{SiO2} = 1.57$, noticeably larger than the refractive index of the $SiO_2$ bulk material (1.45 at 780 nm). From this analysis, we conclude that our cavity sensor is sensitive to the hydrodynamic particle size[13]. With the above results, we can calculate the effective refractive index of the hydration shell to be $n_h = 1.39 \pm 0.04$. Analogous evaluations for sample A yield a hydration shell with 8.5 nm thickness and $n_h = 1.40 \pm 0.02$. This analysis shows that our measurement technique allows us to quantify properties of the hydration shell.

**Particle tracking**. As a next step, we introduce a method to achieve 3D tracking of nanoparticle motion within the cavity. Therefore, we make use of the signal of two higher-order transverse modes in addition to the fundamental cavity mode. To reduce the required amplitude for the cavity length modulation, we use two lasers with different wavelengths such that the fundamental mode probed by laser 1 appears in the middle of the frequency-split $TEM_{01}/TEM_{10}$ modes which are probed by laser 2. Figure 3a shows a cavity transmission measurement of the three modes together with their lateral intensity distributions. The modes have complementary spatial distributions, leading to a unique set of corresponding frequency shifts produced by a nanoparticle at any given position inside the light field. Due to symmetry, this uniqueness is limited to one octant of a cartesian coordinate system that is centered at one field antinode, such that the assignment of spatial positions will be folded into this

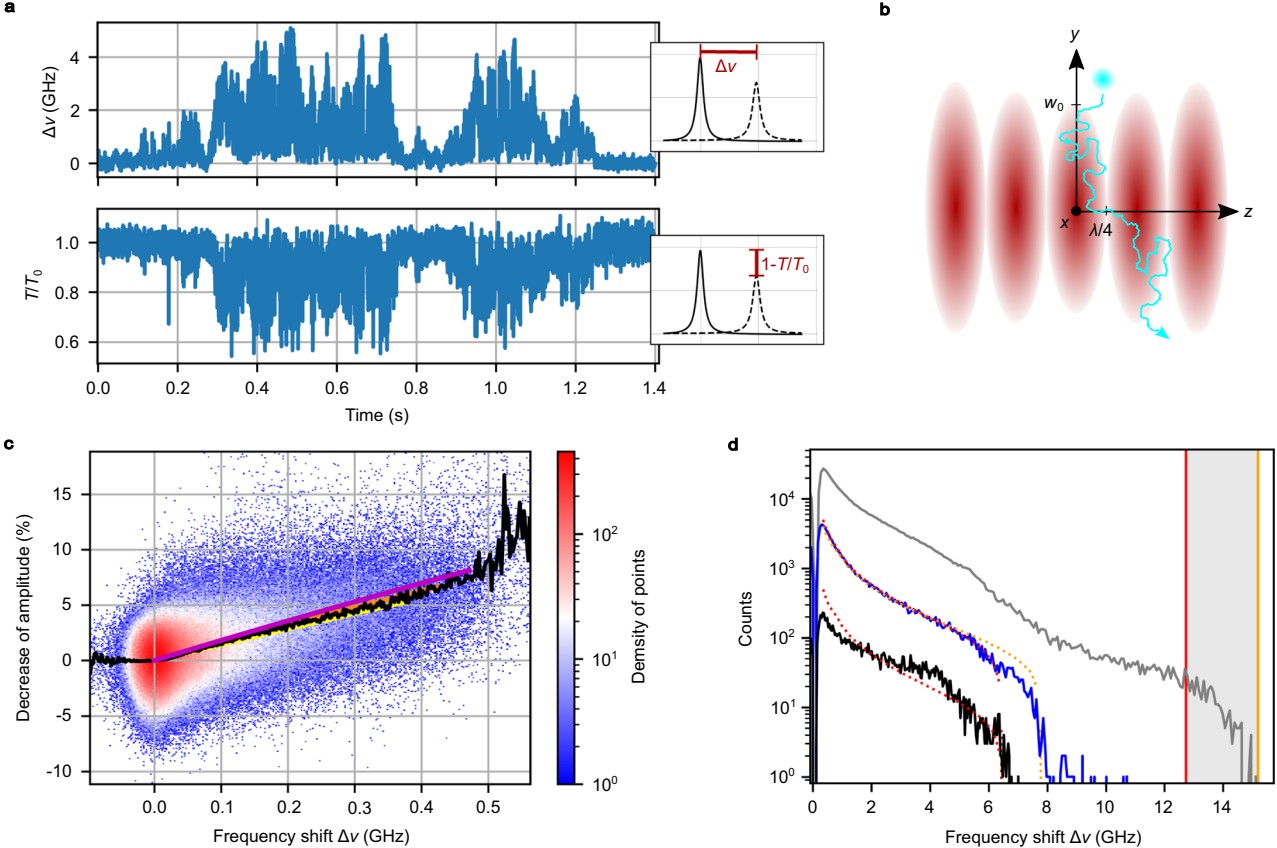

**Fig. 2 Quantitative nanoparticle characterization. a** Cavity frequency shift $\Delta\nu$ and peak transmission amplitude $T/T_0$ as a function of time produced by a single SiO$_2$ nanoparticle. **b** Schematic sketch of a single nanoparticle diffusing through the standing wave cavity field. **c** Correlation of cavity frequency shifts and transmission reductions. Solid black line: mean transmission reduction for a given frequency shift. Solid lines: simulation for the nanoparticle's refractive indices $n_{NP} = \{1.41, 1.42, 1.43\}$ (yellow, orange, purple). **d** Histograms for the measured frequency shifts of a single transit event (black data) also shown in Fig. 3b, 59 transits (blue data, measurement duration: 2 h) and 210 additional transits (gray data, 8 h). The observed increased shifts reveal agglomeration. Dotted lines: simulation for $n_{NP} = 1.43$ and the hydrodynamic radii $r_{hydr} = \{77.4, 82.3\}$ nm (red, orange). The vertical solid red and orange lines depict the corresponding estimated dimer shifts.

domain. However, this does not alter the properties of the inferred Brownian motion, and effects such as anisotropy or reduced dimensionality of diffusion would remain visible.

Using cavity 2, we measured several hundred transit events with the three modes at the same time (see Supplementary Note 1). Figure 3b shows a representative time trace for a single nanoparticle (sample B) entering the cavity mode multiple times (see Supplementary Fig. 13 for detailed evaluation). One can see the different behavior of the three modes depending on the nanoparticle's position inside the cavity light field. In Fig 3c, the corresponding positions are schematically sketched for different times marked in orange in Fig. 3b. The frequency shift distribution of the fundamental mode is shown in Fig. 2d (black line), those for the higher-order modes are shown in Supplementary Fig. 12 (red lines), and all three distributions are consistent with a single nanoparticle. To obtain the most probable nanoparticle position for a given time, we compare the measured frequency shift triples with simulated shifts for all possible nanoparticle positions (see Supplementary Notes 3 and 4). Due to the present measurement noise for each mode (see Supplementary Fig. 14), multiple positions are compatible with the measured shifts from which we select the mean value (see Supplementary Fig. 16). This gives the 3D coordinates of the nanoparticle for each measurement time (see Fig. 3d). Figure 3e shows the nanoparticle movement in the 3D space for the time interval marked in gray in Fig. 3b, d. The signal-to-noise ratio (SNR) is

the quantity that determines the uncertainty of the localization. In the current measurement, we obtain SNR $= \Delta\nu_{00,\text{max}}/\sigma_{00} = 53$, where $\Delta\nu_{00,\text{max}}$ is the maximal frequency shift at the center of the TEM$_{00}$ antinode and $\sigma_{00}$ is the frequency noise of this mode (see Supplementary Fig. 14). Figure 3f shows the calculated localization uncertainties for the measured noise in the $xz$- and $xy$-plane for $y = 0$ and $z = 0$, respectively. We define the localization uncertainty as $\delta r = (\delta x\delta y\delta z)^{1/3}$ from the respective coordinate range that is compatible with the measurement including the noise for a single measurement, i.e. a corresponding time scale of 0.3 ms. The minimal localization uncertainty is found to be 8 nm at the points of largest intensity gradients, and the mean uncertainty within the sensing volume which extends up to the $1/e^2$ contour of the standing wave is 44 nm. This is on par with state-of-the-art nanoparticle localization techniques on such short timescales[4,33–37], but now this is achieved for the localization in three dimensions and with label-free nanoparticles.

We are able to deduce the diffusivity as well as the nanoparticle size from the 3D track by calculating the mean squared displacement (MSD($\tau$)) (see Fig. 3g and Supplementary Fig. 17 for additional information). We perform a linear fit of the MSD data using an optimized number of data points weighted with their respective statistical errors $w = 1/\Delta x$, $\Delta x$ being the uncertainty of the mean, and an $y$-offset originating from noise[38]. This gives a mean diffusivity of $\langle D \rangle = (2.2 \pm 0.3)\ \mu m^2/s$ and a hydrodynamic radius of $r_{hydr} = (96 \pm 15)$ nm. An excellent agreement with the theoretical

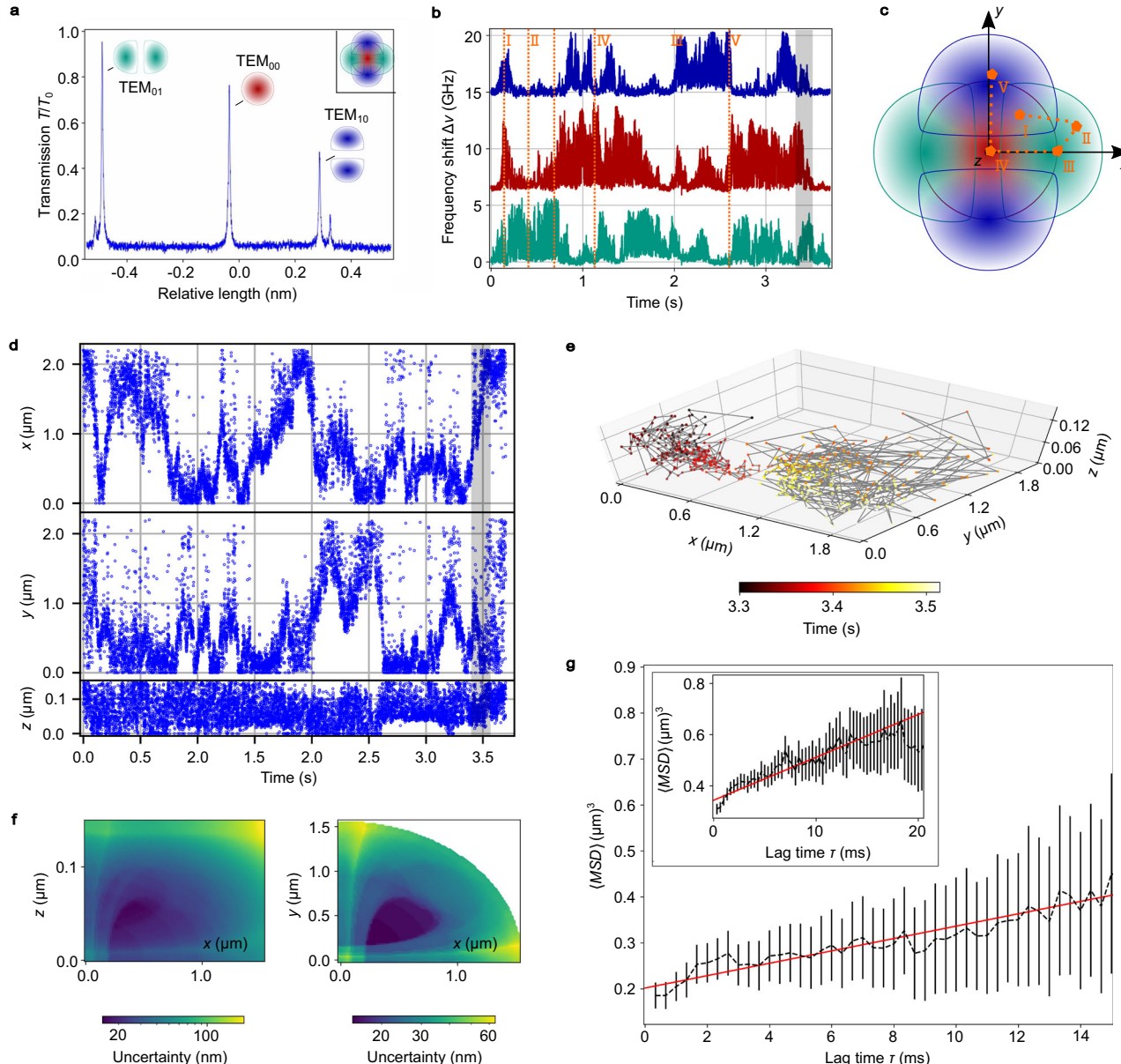

**Fig. 3 Three-dimensional tracking of a single SiO$_2$ nanosphere and determination of its diffusivity. a** Cavity transmission signal $T/T_0$ with the three evaluated transverse modes TEM$_{00}$, TEM$_{01}$, and TEM$_{10}$ indicated. Inset: Schematic illustration of the complementary spatial regions covered by the modes. **b** Time trace of frequency shifts of the three modes. **c** Sketch of particle positions at different times depicted in (**b**). **d** Evaluated positions by taking the mean position at a given time. **e** Three-dimensional representation of the position tracking. **f** Calculated position-dependent localization uncertainty. **g** Mean squared displacement (MSD) for the single transit event shown in **d** and the MSD averaged over five single nanoparticle transit events (inset). The black error bars represent the statistical error (×10 for visibility) and the red line is a weighted linear fit.

value of $D = 2.84$ μm$^2$/s can be achieved by calculating the diffusivity from five single-particle transit events (see inset Fig. 3g). Here we get $\langle D \rangle = (2.8 \pm 0.4)$ μm$^2$/s and $r_{hydr} = (77 \pm 10)$ nm.

As an important result of this analysis, the particle tracking allows us to infer the hydrodynamic radius of a nanoparticle, while from the frequency shift probability distribution (see Fig. 2d) we obtain the polarizability. Measuring the cavity transmission change in addition provides a higher precision and makes the measurement immune against, e.g., systematic errors in the cavity geometry. Combining only these two measurements, which can be performed at the same time, we can deduce the effective refractive index of a nanoparticle.

## Discussion
Our work demonstrates a compact and robust ultra-sensitive microcavity sensor that can be produced in a controlled and

reproducible process. It has shown stable operation over several weeks in a given configuration and is thus useful for extensive studies. We have introduced the simultaneous measurement of several cavity transverse modes to achieve 3D particle tracking inside the cavity, which is an important step to combine the exceptional sensitivity of cavity-based sensors with spatial imaging. We have used this to demonstrate the quantitative characterization of the nanoparticle size and effective refractive index. The noise level of the current measurements would allow the investigation of particles with hydrodynamic radii down to 20 nm. Significant further improvements in sensitivity are expected by active stabilization of the cavity, smaller mode volumes, and noise-rejection techniques such as heterodyne spectroscopy, such that few-nm large biomolecules can be expected to become detectable. Increased SNR and further optimization, e.g., with Bayesian analysis[6] could offer improved precision for position

tracking. Using two-color measurements of the cavity fundamental mode would enable the extension of the z-sensing volume over a large fraction of the cavity volume[39], without compromising the spatial resolution, and schemes based on frequency shifts of a degenerate higher-order cavity mode family could resolve the sign-ambiguity in the x, z-plane such that also the full mode cross-section could be used[26]. Operating under active stabilization will allow to harness the full bandwidth of the sensor, which in principle is bounded only by the cavity decay rate of ~$10^8$ Hz. This could enable the label-free study of dynamical processes of biologically relevant processes such as protein folding, dynamics of drug release from nano-carriers, or molecular binding on a single-particle level.

## Methods

**Experimental set-up**. Our measurements were taken with two different cavity configurations (cavity 1 and 2). The following parameters were determined for the cavities filled with water. Cavity 1 had a finesse of 18,400, an effective cavity length of 27.3 μm and the mirrors had a mean central radius of curvature of $r_{c,SM} = 59.3$ μm and $r_{c,MM} = 94.5$ μm. Cavity 2 had a finesse of 56,710 at the effective cavity length of 5.4 μm. Here we used mirrors with $r_{c,SM} = 49.6$ μm and $r_{c,MM} = 83.5$ μm. The effective cavity length $d = d_g + d_p$ in water is given by the mirror separation $d_g$ and the field penetration length $d_p$ which can be as small as $d_p = 430$ nm and defines together with the radius of curvature the mode radius $w_0 = 2.3$ μm and the mode volume $V_m = 110.0$ μm³ for cavity 1 and $w_0 = 1.5$ μm and $V_m = 10.0$ μm³ for cavity 2. The quality factor depends on the mirror separation. Here it is $Q = 1.7 \times 10^6$ for cavity 1 and $Q = 1.0 \times 10^6$ for cavity 2. In the quantitative measurements, we use a grating-stabilized diode laser at 780 nm to probe the cavity, and detect the transmitted light by a photodetector (see Supplementary Fig. 1). We modulate the cavity length at a frequency of 3–10 kHz with an amplitude of 150 pm to repeatedly sample the cavity fundamental mode.

**Data analysis**. The high-frequency signals are recorded by an oscilloscope which is operated in sequence mode. From the resonance curve of each trigger event the amplitude and the time position of the amplitude in relation to the trigger is extracted (see Supplementary Fig. 4). In order to remove cavity drifts (mainly thermal expansion) from the frequency shift data, we subtract parabolas fitted to the background noise (see Supplementary Fig. 5).

**Nanoparticles**. We use silicon dioxide nanoparticles from two different companies: SiO₂ spheres with a hydrodynamic radius of $r_{hydr} = 71.5$ nm, a standard deviation of $\sigma = 2$ nm, and a refractive index of $n_{SiO2} = 1.42$ from microparticles GmbH (sample A), and SiO₂ spheres with a geometric radius of $r = 60$ nm and $\sigma < 2.2$ nm from ALPHA Nanotech (sample B). More information can be taken from Supplementary Figs. 9 and 10.

## Data availability

The data presented in this publication are available from the corresponding authors on request.

## Code availability

The codes are available from the corresponding authors on request.

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

## Acknowledgements

This work was partially funded by the Deutsche Forschungsgemeinschaft (DFG), the Cluster of Excellence Nanosystems Initiative Munich (NIM), and the Karlsruhe School of Optics and Photonics (KSOP). It was supported by the Laser Material Processing (LMP) infrastructure of the Karlsruhe Nano Micro Facility (KNMF), a Helmholtz Research Infrastructure at Karlsruhe Institute of Technology (KIT). We thank L. Radtke, P. Brenner, M. Schumann, and M. Rutschmann for experimental assistance and T. Hümmer, M. Hippler, and T. W. Hänsch for helpful discussions.

## Author contributions

L.K., M.M. and D.H. conceived the original idea, L.K. performed theoretical calculations, L.K. and C.K. prepared the experimental set-up, L.K. performed the experiments and analyzed the data, L.K., M.M., C.K., M.W. and D.H. discussed the results, L.K. and D.H. wrote the manuscript, and M.M., C.K. and M.W. provided feedback.

## Funding

## Competing interests

The authors declare no competing interests.
