## [Peer Review File · Nature Communications]

REVIEWER COMMENTS

Reviewer #1 (Remarks to the Author):

This manuscript reports a Fabry-Perot cavity based system for three dimensional trapping and characterization of single nanoparticles. These sorts of tool are important for a wide variety of research areas, from molecular dynamics to material science and air quality measurements. As such, they have been studied quite extensively, though new innovations are regularly made.

As I see it, the primary novelty of this work is the use of three cavity spatial modes to allow three-dimensional localization of the particles. The other results of the paper are in my view, relative to this, quite incremental. Overall, however, I did not find these results especially convincing, and probably not of the novelty required for a journal such as Nature Communications.

Some specific issues:

1) The authors quote spatial and temporal resolutions of around 8 nm and 1 ms. However, they don't put this in context. optical tweezers, for example, allow three dimensional particle tracking and have far better spatial resolution down to around 10 fm. Of course, they do this with larger particles. Never-the-less a careful comparison of how good the particle tracking achieved here is compared to the state of the art would clarify how significant the results may be.

2) The particle tracking data is extremely unconvincing, especially in the z-axis, where the time trace is wildly oscillating, and in the three dimensional trace (fig 3e) that I struggle to learn anything meaningful from. Why is the diffusion in the z axis so much faster than in x and y? The MSD generated from the x-y data is also unconvincing. Why does this oscillate so much? It's especially unusual to see an MSD decrease with increasing wait time. That suggests some correlations over time in the particle position. Can the authors explain why such corrections would exist?

3) Also related to the particle tracking data, I find it somewhat suspect to, when deciding on which position to attribute a particle at a given time, to require this to be the one that is closest to the position at the previous time step. In the limit of large noise, this would lead to the erroneous conclusion that the particles motion was frozen. Surely the statistically appropriate way to to this would be to find the position that minimizes the mean-squared error in the fit to the parameters measured, and to use the region over which the mean-squared error is within a factor of two of this minimum to determine the uncertainties in the particle position?

4) In the introduction the authors state that previous work that provides spatial information about the tracked particle include scanned microcavities and point out problems with this, implying that

their manuscript takes a different approach. But in fact they do use scanning, just with two lasers. Clearer arguments would be appreciated here.

5) The authors state that their cavities have Q/V similar to WGM approaches, but don't make the explicit comparison. They should do this.

6) When calculating the refractive index of 1.41 for sample A, the authors appear to assume that the hydration layer has the same refractive index as silica (while later calculating it to be different). Can this be justified?

7) The authors "notably" observe that the transmission change and frequency shift of the cavity change in the same way with modeshape, so that the ratio is independent of this. But this is well known. See e.g. Zhu et al Nature Photonics 4 46 (2010).

8) The authors conclude that their experiments are sensitive to hydrodynamic particle size. This can be easily tested more directly by introducing salt into the solution, as in their reference [12].

9) On page 4, the authors state an $SNR=51$, but don't define how they calculate it.

Reviewer #2 (Remarks to the Author):

The authors report on a method for tracking and characterising nanoparticles based on measuring the time dependent frequency shifts of multiple cavity modes within an open optical microcavity formed between the ends of a pair of optical fibres. The work builds on previous methods for nanoparticle characterisation within an optical microcavity.

The manuscript is very clearly written, the experimental methodology and data analysis is sound, and the work represents a significant advance in the field of nanoparticle characterisation. I would be very happy to see the work published in Nature Communications. I have a few comments that the authors might like to consider in preparing a final version of their manuscript.

1. In the introduction, it would be worth explaining for the relatively broad audience of Nature Communications readers why nanoparticle tracking and characterisation are important. What new applications will open up or be facilitated as a result of the methodology developed by the authors? What areas of science will it impact? The first sentence of the abstract states that the dynamics of nanosystems are very important, but I think some specific examples would be appreciated by the reader.

2. There are a number of places where variables appearing in equations or in the text are not defined. The authors should ensure that all variables are properly defined the first time they appear.

3. On page 4, the authors suggest that “continuous adsorption of molecular layers on the nanoparticles” might explain the larger frequency shifts in the cavity modes that are seen at longer times. I may have missed something in the description of the sample preparation, but I am mystified by this proposal. If I have understood correctly, the sample comprises silica nanoparticles suspended in doubly distilled water. What are the adsorbing molecules that are proposed? Water, or something else? I would expect that the solvation structure around the nanoparticles would be established very rapidly, certainly on a timescale of much less than a second. Unless some charge-related property of the nanoparticles is changing drastically over the timescale of hours, I see no reason why the structure of the solvation shell should change significantly. Agglomeration of nanoparticles is a much more plausible explanation for the observed behaviour of the frequency shifts. This explanation could be tested by introducing a stabilising agent to eliminate (or at least reduce) agglomeration, and observing whether or not the time-dependent behaviour of the frequency shifts persists.

4. Could the symmetry of the cavity modes be broken by using a differently shaped cavity, in order to achieve truly three-dimensional localisation of the particle, or is there always an eight-fold degeneracy in the position measurement?

5. Caption to Figure 3(e). I think “two-dimensional representation of the position tracking” is a confusing description for a plot of a three-dimensional track. “Three dimensional plot of the position tracking” would be clearer.

6. Page 5, line 318. When stating a measured value together with its uncertainty, the convention is usually to report the uncertainty to one (or at most two) significant figures, and to round the value to match, i.e. (87 ± 24) nm would seem a more sensible way to report the measured hydrodynamic radius than (86.9 ± 23.6) nm, i.e. the decimal place in the value is somewhat meaningless when there's a ± 24 nm uncertainty.

7. How reproducible is the fabrication of the microcavities and overall setup? Is it possible to make identical set ups, or will each one have different mirror radii of curvature, finesse, and mode characteristics, i.e. is each device a ‘one-off’, requiring extensive calibration and characterisation, or is there potential to develop the method into a new general approach to nanoparticle characterisation?

Minor typographical errors, etc

1. First line of abstract, ‘contains’ should be ‘contain’.

2. Sixth line of abstract, 'transversal' should be 'transverse'.
3. Page 2, line 117, 'Finesse' does not need to be capitalised.
4. Page 2, line 161, 'electrical' should be 'electric'.
5. Page 4, line 226. It would be clearer to write "For the calculation of the refractive index from the data shown in Fig 2d...".
6. Page 5, line 317. It would be clearer to write $(2.5 \pm 0.7) \times 10^{-12} \text{ m}^2 \text{ s}^{-1}$.

Reviewer #3 (Remarks to the Author):

In this paper, Kohler et. al. have used a Fabry-Perot cavity formed by two mirror-coated optical fiber facets to detect nanoparticles (through the frequency shifts they induce) and to achieve 3D tracking of the motion of particles (by monitoring the changes on three transversal modes of the cavity). This open cavity configuration has allowed them to detect SiO₂ particles down to 60 nm with 300 microsecond temporal and 8 nm spatial resolution.

Open-cavity configurations to detect particles and measure their refractive index and size have been previously used by monitoring single or multiple cavity resonances have been used in different implementations in works as cited by the authors (e.g., Nano Lett. 2016, 16, 6172–6177, Lab Chip, 2014, 14, 4244–4249). Fiber-based Fabry-Perot resonators at the facets of fibers have been well-established and been in use in the studies of cavity-QED and quantum optomechanics.

The novelty of this manuscript lies in the use of a Fabry-Perot cavity formed at the facets of two optical fibers in microfluidic settings for particle detection and tracking. By modulating the cavity length, the fundamental cavity resonance could be repeatedly sampled to monitor the change in the resonance frequency and the amplitude of the cavity transmission.

The experiments have been well-prepared and executed with sufficient rigor. The manuscript is well-written and the end-results and all the required information to understand the methods is given. The motivation behind such a study is also justified.

I find the content and end-results of this manuscript to be very interesting and significant. The techniques developed in this manuscript will be of great interest for researchers in the field of single particle, including biomolecules, tracking and characterization. I think the manuscript deserves publication. The authors should clarify the following in a revised manuscript:

1. The authors mention single particle tracking and detection. How do the authors ensure that there is one and only one particle within the cavity mode field? What happens when more than one particle enter the mode field?

2. Will the system develop here allow tracking and characterizing multiple particles?

3. What limits the cavity modulation frequency? and How does this frequency affect the measurement outcome?

Response to the reviewers

We thank the reviewers for their critical assessment of our work. In the following we address their concerns point by point.

Reviewer 1

This manuscript reports a Fabry-Pérot cavity based system for three dimensional trapping and characterization of single nanoparticles. These sorts of tool are important for a wide variety of research areas, from molecular dynamics to material science and air quality measurements. As such, they have been studied quite extensively, though new innovations are regularly made.

As I see it, the primary novelty of this work is the use of three cavity spatial modes to allow three-dimensional localization of the particles. The other results of the paper are in my view, relative to this, quite incremental. Overall, however, I did not find these results especially convincing, and probably not of the novelty required for a journal such as Nature Communications.

Reply: We thank the referee for their time in reviewing our manuscript. We agree that particle tracking using multiple cavity modes is the main novelty of our work that enables the detection of Brownian motion at high bandwidth over extended time for the first time in a microcavity. Particle tracking provides significant additional information beyond the discrete binding / unbinding events observed typically e.g. with WGM resonators. But also other aspects, despite more technical, can be considered as significant advances: Our cavity sensor achieves a record sensing figure of merit ($\sim Q/V$) - it has a quality factor as high as WGM cavities but a factor 100x smaller mode volume and correspondingly 100x higher sensitivity compared to WGM biosensing cavities; our cavity has a more than 10x higher cavity finesse for an open-access microcavity in liquid than previous work ($F = 62500$ vs $F \sim 5000$ in Trichet et al. [20], the only other publication reporting single nanoparticle detection in an open-access cavity); we report the first realization with a fully fiber-based cavity; we introduce a novel approach to combine alignment-free, monolithic cavity mounting in a ferrule with 3D laser-printed microfluidic structures into a compact device. Overall, we believe that our device has the potential to achieve improved sensitivity compared to earlier work, which combined with the new capability of cavity-based single particle tracking can enable new insight. Together with the robust and compact design which requires only smallest analyte amounts, it can be of use for a range of applications.

Reviewer Point P 1.1 — The authors quote spatial and temporal resolutions of around 8 nm and 1 ms. However, they don't put this in context. optical tweezers, for example, allow three dimensional particle tracking and have far better spatial resolution down to around 10 fm. Of course, they do this with larger particles. Never-the-less a careful comparison of how good the particle tracking achieved here is compared to the state of the art would clarify how significant the results may be.

Reply: We thank the referee for pointing out these key parameters. Indeed we have missed to put this in context. In our work, we achieve 8 nm spatial resolution within 0.3 ms, without averaging. This is a key requirement for being able to resolve the fast Brownian motion of small nanoparticles, where positions change significantly on a ms time scale. In contrast, the high spatial localization precision mentioned by the referee is achieved by observing trapped particles and averaging over several seconds, such that one can only make a statement about the average position from that, see e.g. Perkins et al.,

Annu. Rev. Biophys 43, 279 (2014). If we compare these results to ours at the short measurement time of 0.3 ms, we obtain an as good resolution, despite studying significantly smaller particles. We note that by using an actively stabilized cavity, we can expect to increase the temporal resolution to below $1\mu\text{s}$ without loss of spatial resolution. In the revised version of our manuscript, we have added a comparison to put our results in context.

Reviewer Point P 1.2 — The particle tracking data is extremely unconvincing, especially in the z-axis, where the time trace is wildly oscillating, and in the three dimensional trace (fig 3e) that I struggle to learn anything meaningful from. Why is the diffusion in the z axis so much faster than in x and y? The MSD generated from the x-y data is also unconvincing. Why does this oscillate so much? It's especially unusual to see an MSD decrease with increasing wait time. That suggests some correlations over time in the particle position. Can the authors explain why such corrections would exist?

Reply: We thank the referee for pointing out this possible misinterpretation. The apparent faster z-motion originates from the different scaling of the axes. As we describe in the manuscript, the resolution of z-positions is limited to half of a standing wave ($\lambda/(4n)$), i.e. a length scale of about 130nm. The shown length scales of the x,y coordinates are thus much larger ($2\mu\text{m}$) than the one for the z coordinate, which makes the z-motion appear different. Since the average particle diffusion distance between subsequent measurement points is on the same order of magnitude as the z-range, it is as expected that the z-coordinate changes over its full reconstruction range within a few measurement points. We note that probing the fundamental mode frequency shifts of two lasers with different frequency would allow us to also resolve z on a much larger length scale and avoid the current limitation. We added a comment to make the origin of the apparent faster z-motion, the current limitation in z reconstruction, and the possible future improvement clear.

Regarding the MSD: Since the MSD is taken from a single transit event, the specific diffusion trajectory can well lead to decreasing MSD values for longer time. E.g. a single particle can happen to diffuse back to its location from a previous time, which would lead to a smaller MSD for certain times. Only when averaging over many transit events, one will be able to see the statistical behavior where the MSD increases linearly on all time scales. We have observed this behaviour also for simulated trajectories and their MSD. To take this better into account, in the revised version of our manuscript, we have fitted the MSD only at early times where there is sufficient statistics to get a reliable value for its slope. Therefore, we have made use of a more careful treatment following Michalet et al., Phys Rev E 82, 041914 (2010), and have chosen the optimal number of data points for the fit, and have weighted the data points with their inverse uncertainty.

Reviewer Point P 1.3 — Also related to the particle tracking data, I find it somewhat suspect too, when deciding on which position to attribute a particle at a given time, to require this to be the one that is closest to the position at the previous time step. In the limit of large noise, this would lead to the erroneous conclusion that the particles motion was frozen. Surely the statistically appropriate way to do this would be to find the position that minimizes the mean-squared error in the fit to the parameters measured, and to use the region over which the mean-squared error is within a factor of two of this minimum to determine the uncertainties in the particle position?

Reply: We thank the referee for bringing up this point and agree that the choice of minimal distance will introduce a bias. Our reasoning was that as long as the noise is not fully masking the signal,

one will still be able to follow the trajectory as soon as it deviates from its previous value by more than the noise. This means that there will be times where the particle is "frozen", but then it will jump to a new position as soon as we can deduce it from the measurement with enough certainty. To analyze this effect in more detail, we have performed Monte Carlo Simulations of particle trajectories, calculated the corresponding cavity mode frequency shifts, added noise, and fed the "noisy data" in our algorithm to find the reconstructed trajectory. From this we found that the "minimal-distance" approach is indeed slightly worse than choosing the mean value of the compatible positions. We have modified our algorithm and now use the mean position. This also improves the MSD data.

Reviewer Point P 1.4 — In the introduction the authors state that previous work that provides spatial information about the tracked particle include scanned microcavities and point out problems with this, implying that their manuscript takes a different approach. But in fact they do use scanning, just with two lasers. Clearer arguments would be appreciated here.

Reply: Indeed our formulation was not concise enough to make the desired point clear. We wanted to refer to earlier work based on so-called scanning cavity microscopy, where the cavity is scanned across the sample in a lateral manner, similar to an AFM or a confocal microscope, and scanning was meant to refer to lateral spatial scanning. Here, we have a fixed cavity, but indeed, we still scan the cavity length. We have reformulated the respective sentence to differentiate this aspect.

Reviewer Point P 1.5 — The authors state that their cavities have Q/V similar to WGM approaches, but don't make the explicit comparison. They should do this.

Reply: We thank the referee for their suggestion and now add a comparison. We achieve for the cavity as used in the experiment in water $Q = 1 \times 10^6$, $V_m = 40(\lambda/n)^3$, $Q/V_m(\lambda/n)^3 = 2.8 \times 10^4$, while microspheres in water achieve $Q \approx 10^6$ and $V_m \approx 10^4(\lambda/n)^3$, yielding $Q/V_m(\lambda/n)^3 \approx 200$ (see e.g. Vollmer, Keng PNAS (2008), Su, Stoltz LSA (2016)). This means, our cavity improves the key figure of merit by a factor > 100 compared to the state-of-the-art WGM biosensors. In the revised version of our manuscript, we have added this comparison.

Reviewer Point P 1.6 — When calculating the refractive index of 1.41 for sample A, the authors appear to assume that the hydration layer has the same refractive index as silica (while later calculating it to be different). Can this be justified?

Reply: We thank the referee for this valuable comment. It is true that we were a bit imprecise: When we calculate the refractive index of 1.41, the effective refractive index is determined, which includes the particle and its hydration shell, instead of the intrinsic refractive index of the bare particle. In comparison, 1.42 is the bare particle refractive index given by the manufacturer. Therefore, in the revised version of our manuscript, we have introduced the effective refractive index earlier in order to avoid any imprecision and have changed the respective parts.

Reviewer Point P 1.7 — The authors "notably" observe that the transmission change and frequency shift of the cavity change in the same way with modeshape, so that the ratio is independent of this. But this is well known. See e.g. Zhu et al Nature Photonics 4 46 (2010).

Reply: We have omitted the word "notably" to avoid the impression that this was our finding and cite Zhu et al. in this context.

Reviewer Point P 1.8 — The authors conclude that their experiments are sensitive to hydrodynamic particle size. This can be easily tested more directly by introducing salt into the solution, as in their reference [12].

Reply: We thank the referee for this comment. Indeed, changing the salt concentration would be an interesting aspect to study, but requires a systematic and carefully conducted measurement series which would go beyond the scope of this manuscript and could rather be content of subsequent work. To evidence the effect here, we have studied two nanoparticle samples with slightly different properties and very similar intrinsic size (sample A: $r_{\text{geom}} = 60$ nm vs sample B: $r_{\text{geom}} = 63$ nm), and we have observed significant differences in their hydrodynamic shell (sample A: $r_{\text{h}} = 8.5$ nm, sample B: $r_{\text{h}} = 15.3$ nm). We have two independent evidences of being sensitive to the hydrodynamic size of the nanoparticle: First, the polarizability of sample B would correspond to an unphysically large refractive index of the nanoparticle if no hydration shell is assumed: if $r_{\text{geom}} = 60$ nm ($r_{\text{hydr}} = 75.3$) the refractive index corresponds to $1.57 \gg 1.45$ ($1.46 \approx 1.45$). The geometric radius was determined by TEM-measurements and the hydrodynamic radius was determined by DLS measurements (see Supplementary Fig. 10). Second, regardless of the refractive index, we could deduce the hydrodynamic radius of sample B from the three-dimensional track. The result ($r_{\text{hydr}} = 76.9 \pm 10.0$ nm) agrees very well with the hydrodynamic radius, which was determined by the DLS measurement ($r_{\text{hydr}} = 75.3$) and differs from the geometric radius ($r_{\text{geom}} = 60$ nm). All in all, we believe this is sufficient evidence for the capability of the cavity sensor to quantify the presence of the hydration shell for a proof-of-principle demonstration.

Reviewer Point P 1.9 — On page 4, the authors state an SNR=51, but don't define how they calculate it.

Reply: We thank the referee for this comment. We calculated the SNR by dividing the maximal measured frequency shift value of the TEM₀₀-mode $\Delta\nu_{00,\text{max}}$ through the noise (see Supplementary Figure 14):

$$SNR = \frac{\Delta\nu_{00,\text{max}}}{\sigma_{00}} = 53 \quad (1)$$

Indeed, when we recalculated this value, we even got a better value for the SNR. We changed the value for the SNR in the manuscript to 53 and added the additional information of how we calculated it.

Reply: The comments of the referee have been particularly helpful to improve our manuscript. We hope that our responses and the clarifications in the manuscript are sufficiently compelling that the referee is willing to reconsider their recommendation for publication in Nature Communications.

Reviewer 2

The authors report on a method for tracking and characterising nanoparticles based on measuring the time dependent frequency shifts of multiple cavity modes within an open optical microcavity formed between the ends of a pair of optical fibres. The work builds on previous methods for nanoparticle characterisation within an optical microcavity. The manuscript is very clearly written, the experimental methodology and data analysis is sound, and the work represents a significant advance in the field of nanoparticle characterisation. I would be very happy to see the work published in Nature Communications. I have a few comments that the authors might like to consider in preparing a final version of their manuscript.

Reply: We thank the referee for this very positive overall assessment.

Reviewer Point P 2.1 — In the introduction, it would be worth explaining for the relatively broad audience of Nature Communications readers why nanoparticle tracking and characterisation are important. What new applications will open up or be facilitated as a result of the methodology developed by the authors? What areas of science will it impact? The first sentence of the abstract states that the dynamics of nanosystems are very important, but I think some specific examples would be appreciated by the reader.

Reply: We have taken this helpful suggestion into account and added application examples in the first paragraph.

Reviewer Point P 2.2 — There are a number of places where variables appearing in equations or in the text are not defined. The authors should ensure that all variables are properly defined the first time they appear.

Reply: We thank the referee for this comment and added missing definitions of the variables in the manuscript.

Reviewer Point P 2.3 — On page 4, the authors suggest that “continuous adsorption of molecular layers on the nanoparticles” might explain the larger frequency shifts in the cavity modes that are seen at longer times. I may have missed something in the description of the sample preparation, but I am mystified by this proposal. If I have understood correctly, the sample comprises silica nanoparticles suspended in doubly distilled water. What are the adsorbing molecules that are proposed? Water, or something else? I would expect that the solvation structure around the nanoparticles would be established very rapidly, certainly on a timescale of much less than a second. Unless some charge-related property of the nanoparticles is changing drastically over the timescale of hours, I see no reason why the structure of the solvation shell should change significantly. Agglomeration of nanoparticles is a much more plausible explanation for the observed behaviour of the frequency shifts. This explanation could be tested by introducing a stabilising agent to eliminate (or at least reduce) agglomeration, and observing whether or not the time-dependent behaviour of the frequency shifts persists.

Reply: We agree that the hypothesis of adsorption of molecular layers is somewhat speculative. Our picture is that residual contamination of the water, which may arise from organic molecules adsorbed to the microfluidic channels, may lead to accumulation of adsorbates on the nanoparticles. However, we have no evidence of such contamination except for increased frequency shifts. On the other hand, we have clear evidence of agglomeration; e.g. the maximum observed frequency shift after a certain time corresponds exactly to two nanospheres. The density would be too low to have a noticeable fraction of more than two agglomerated particles. We thus omit the mentioning of the speculation on adsorption.

Reviewer Point P 2.4 — Could the symmetry of the cavity modes be broken by using a differently shaped cavity, in order to achieve truly three-dimensional localisation of the particle, or is there always an eight-fold degeneracy in the position measurement?

Reply: It is an interesting suggestion to consider cavity modes with broken symmetry to remove the degeneracy. With strongly non-symmetric mirror profiles, e.g. via asymmetric multi-well potentials,

this might be possible. Another approach is to use highly symmetric mirror profiles that allow (near-) degenerate higher-order mode families. The degeneracy would then be broken by the particle, and phase-sensitive interference of the different transverse modes would then yield full lateral position information. This was proposed in Horak et al., PRL 88, 043601 (2002) for a single atom in a cavity. Along the z-coordinate, using two lasers of different frequency will lead to a beating pattern in the cavity, such that by probing the two fundamental mode frequency shifts, one could resolve positions along the half-period of the beating pattern, which can be the entire cavity length. In the revised version of our manuscript, we have mentioned these possibilities in the outlook.

Reviewer Point P 2.5 — Caption to Figure 3(e). I think “two-dimensional representation of the position tracking” is a confusing description for a plot of a three-dimensional track. “Three dimensional plot of the position tracking” would be clearer.

Reply: We thank the referee for drawing our attention to this wrong denotation and changed it correspondingly.

Reviewer Point P 2.6 — Page 5, line 318. When stating a measured value together with its uncertainty, the convention is usually to report the uncertainty to one (or at most two) significant figures, and to round the value to match, i.e. (87 ± 24) nm would seem a more sensible way to report the measured hydrodynamic radius than (86.9 ± 23.6) nm, i.e. the decimal place in the value is somewhat meaningless when there’s a ± 24 nm uncertainty.

Reply: We thank the referee for this comment and report the uncertainty in our revised manuscript as they recommended.

Reviewer Point P 2.7 — How reproducible is the fabrication of the microcavities and overall setup? Is it possible to make identical set ups, or will each one have different mirror radii of curvature, finesse, and mode characteristics, i.e. is each device a ‘one-off’, requiring extensive calibration and characterisation, or is there potential to develop the method into a new general approach to nanoparticle characterisation?

Reply: The fabrication of microcavities has become a very controlled and reproducible process. Regarding the mirror radii of curvature, we achieve an average deviation of fabricated structures of $< 5\%$ from target values. The quality of the mirror coating is dominated by the coating process, and we use a commercial supplier which achieves excellent coating quality within narrow specifications. As a result, finesse values achieve their target value to about 10%. The self-aligning mount with ferrules minimizes preparation alignment efforts, and allows one to exchange cavity fibers. We are thus convinced that the approach is useful for applications and wider use. In the revised version of our manuscript, we have mentioned this potential more explicitly in the outlook.

Minor

Reviewer Point P 2.8 — First line of abstract, ‘contains’ should be ‘contain’.

Reply: We have corrected this typo.

Reviewer Point P 2.9 — Sixth line of abstract, ‘transversal’ should be ‘transverse’.

Reply: We have corrected this typo.

Reviewer Point P 2.10 — Page 2, line 117, ‘Finesse’ does not need to be capitalized.

Reply: We have corrected this typo.

Reviewer Point P 2.11 — Page 2, line 161, ‘electrical’ should be ‘electric’.

Reply: We have corrected this typo.

Reviewer Point P 2.12 — Page 4, line 226. It would be clearer to write “For the calculation of the refractive index from the data shown in Fig 2d...”.

Reply: We have corrected this typo.

Reviewer Point P 2.13 — Page 5, line 317. It would be clearer to write $(2.5 \pm 0.7) \times 10^{-12} \text{ m}^2 \text{ s}^{-1}$.

Reply: We have revised this expression accordingly.

Reviewer 3

In this paper, Kohler et. al. have used a Fabry-Perot cavity formed by two mirror-coated optical fiber facets to detect nanoparticles (through the frequency shifts they induce) and to achieve 3D tracking of the motion of particles (by monitoring the changes on three transversal modes of the cavity). This open cavity configuration has allowed them to detect SiO₂ particles down to 60 nm with 300 microsecond temporal and 8 nm spatial resolution.

Open-cavity configurations to detect particles and measure their refractive index and size have been previously used by monitoring single or multiple cavity resonances have been used in different implementations in works as cited by the authors (e.g., Nano Lett. 2016, 16, 6172 - 6177, Lab Chip, 2014, 14, 4244 – 4249). Fiber-based Fabry-Perot resonators at the facets of fibers have been well-established and been in use in the studies of cavity-QED and quantum optomechanics.

The novelty of this manuscript lies in the use of a Fabry-Perot cavity formed at the facets of two optical fibers in microfluidic settings for particle detection and tracking. By modulating the cavity length, the fundamental cavity resonance could be repeatedly sampled to monitor the change in the resonance frequency and the amplitude of the cavity transmission.

The experiments have been well-prepared and executed with sufficient rigor. The manuscript is well-written and the end-results and all the required information to understand the methods is given. The motivation behind such a study is also justified.

I find the content and end-results of this manuscript to be very interesting and significant. The techniques developed in this manuscript will be of great interest for researchers in the field of single particle, including biomolecules, tracking and characterization. I think the manuscript deserves publication. The authors should clarify the following in a revised manuscript:

Reply: We thank the referee for this very detailed and positive overall assessment of our manuscript.

Reviewer Point P 3.1 — The authors mention single particle tracking and detection. How do the authors ensure that there is one and only one particle within the cavity mode field? What happens when more than one particle enter the mode field?

Reply: We prepare a strongly diluted solution such that most of the time, no particle is within the cavity volume, and about one particle within 10s is observed. This reduces the probability of having more than one particle in the sensing volume. For instance at the 3D track measurements, about 86% of the measured transit events have been identified as single nanoparticles within the first hour of measurement time. In the measurement, we analyze the frequency shift and transmission change of the cavity resonances. From their correlation and in particular from the observed maximal frequency shift (fig 2e,d), we can confirm whether a single particle is present or several particles. The shape of the frequency shift distribution (Fig. 2d) gives a particularly clear signature. However, for an individual event, in particular for short transits, there can be rare cases where the simultaneous presence of two particles might remain unnoticed. If two or more particles are present, larger maximal frequency shifts and transmission reductions will be very well visible (see Fig. 2 d - black data is from a single particle, magenta for an agglomerate consistent with two particles), and contributions to the frequency shift distribution which are not possible for single particles. In the revised version of our manuscript, we have described more explicitly how we ensure that predominantly single particle events can be seen and how we use the data of Fig. 2c,d to resolve single particle events.

Reviewer Point P 3.2 — Will the system develop here allow tracking and characterizing multiple particles?

Reply: At this point, we do not see how a single cavity could be useful for tracking multiple particles. However, arrays of micro cavities have been demonstrated (e.g. 100 cavities in Wachter et al., Light:Science and Applications 8,37 (2019)), which may allow for parallel multi-particle tracking.

Reviewer Point P 3.3 — What limits the cavity modulation frequency? and How does this frequency affect the measurement outcome?

Reply: The modulation frequency is limited by the dynamic range of the piezo and the excitation of mechanical vibrations of the setup that reduce the frequency stability of the cavity. We have performed experiments up to 14kHz (7kHz) modulation frequency with scanning amplitudes of 0.2nm (1nm). The values were limited by demanding no significant increase of the cavity frequency noise. When even smaller modulation amplitudes can be used, also higher frequencies are possible: In a different setup, values up to 500kHz have been possible for modulation amplitudes of ~ 20 pm without introducing excessive mechanical noise. Higher frequencies will enable a better temporal resolution of the Brownian motion, which is important for smaller objects. On the other hand, with too slow modulation one would miss most of the fast Brownian motion.

Increasing the detection bandwidth to the nanosecond regime will be possible by changing to an actively stabilized cavity, which will be subject of future work. This is mentioned in the Outlook.

REVIEWER COMMENTS

Reviewer #1 (Remarks to the Author):

The authors have made some significant improvements in response to my review and those of the other referees. For instance, I am pleased to see that they are now perform a more rigorous and careful calculation of the mean squared displacement, and that they have removed the bias in their particle tracking algorithm (assuming that the new particle position is the same as the previous one, as long as this is within uncertainties). I am also comfortable with their tests of the sensitivity to the hydrodynamic radius (though adding salt to the solution would be a much clearer test).

Overall, however, I remain of the view that the work is probably suited to a more specialized journal. This is not to say that the authors have not achieved some significant technical developments, just that they have not in my opinion demonstrated a sufficiently significant application or advance in absolute terms (e.g. in sensitivity) as to warrant Nature Communications.

Some of the changes the authors have made to the manuscript have gone backwards. For instance:

* their new claim that their Q/V is 100 times superior to the previous state-of-the-art for WGM resonators is erroneous. Microtoroid based single nanoparticle and biomolecule sensing experiments achieve Q's above 10^8 (100 times better than the experiments that the authors refer to as the state-of-the-art), with mode volumes smaller than those of microspheres as well (so Q/V more than 100 times better than microspheres). There are many papers that use these microtoroid sensors (e.g. PNAS 111 14657 (2014)). The authors need to be more careful here.

* Similarly, the authors now briefly compare their work to localisation microscopy, but not to the state-of-the-art there. Their conclusion is that they achieve comparable sensitivity to the papers they compare with. This is concerning: localisation microscopy is simple and easy to apply, and can give three dimensional information. No cavity is required, the approach can be applied in a conventional microscope, and it does not suffer from the dynamic range issues problematic here. The conclusion then is that, with all the additional effort and challenges, the current paper does not achieve better performance than easier to implement, widely used, techniques that suffer much fewer drawbacks. This illustrates why, in my view, the current work is not quite there for Nature Communications. Were it to show a substantial improvement in sensitivity over the state of the art, then things would get interesting.

I would reiterate from my previous review that optical tweezers based particle tracking can achieve far higher sensitivity than reported here (and does not require the averaging over seconds stated by the authors in their response). For instance [Chavez, Rev. Sci. Inst. 79 105104 (2008)] has around 10 fm/rtHz sensitivity, which corresponds to 0.6 picometers over the authors 0.3 ms measurement time. Clearly this is far superior to what the authors report. Of course, this is achieved for much larger particles. Still, if the authors wish to make comparisons with the state-of-the-art (which I think is really needed for a paper like this), I would encourage them to be comprehensive, rather than picking only a limited set of particular comparisons.

Reviewer #2 (Remarks to the Author):

The authors have addressed all of my comments in the revised version of the manuscript, and I am therefore very happy for the manuscript to be published in Nature Communications.

Reviewer #3 (Remarks to the Author):

The authors have done a great job in addressing the comments and concerns of the Reviewers in the revised version of the manuscript. I think the techniques developed in this manuscript and the demonstrated performance of the system have the potential to find widespread use in label-free detection and characterization of particles as well as the label-free study of the dynamics of biologically relevant particles and processes.

The paper is well-written, Supplement provides sufficient information to understand both experimental and numerical techniques used in the assessment of the performance of the device. In short, I am happy with the revised version of the manuscript and can recommend it for publication in Nature Communications.

Response to the reviewers

We thank the reviewers for spending their time to go through our manuscript and we are glad that all reviewers are satisfied with our corrections and additional considerations of the 3D tracking algorithm and MSD evaluation. Only reviewer one still has doubts if our work is significant enough for the publication in Nature Communications. We want to thank referee two and three for their recommendation of our work being published in Nature Communications. In the following we address the concerns of reviewer one.

Reviewer 1

The authors have made some significant improvements in response to my review and those of the other referees. For instance, I am pleased to see that they are now perform a more rigorous and careful calculation of the mean squared displacement, and that they have removed the bias in their particle tracking algorithm (assuming that the new particle position is the same as the previous one, as long as this is within uncertainties). I am also comfortable with their tests of the sensitivity to the hydrodynamic radius (though adding salt to the solution would be a much clearer test).

Overall, however, I remain of the view that the work is probably suited to a more specialized journal. This is not to say that the authors have not achieved some significant technical developments, just that they have not in my opinion demonstrated a sufficiently significant application or advance in absolute terms (e.g. in sensitivity) as to warrant Nature Communications.

Q/V sensitivity

Some of the changes the authors have made to the manuscript have gone backwards. For instance: * their new claim that their Q/V is 100 times superior to the previous state-of-the-art for WGM resonators is erroneous. Microtoroid based single nanoparticle and biomolecule sensing experiments achieve Q 's above 10^8 (100 times better than the experiments that the authors refer to as the state-of-the-art), with mode volumes smaller than those of microspheres as well (so Q/V more than 100 times better than microspheres). There are many papers that use these microtoroid sensors (e.g. PNAS 111 14657 (2014)). The authors need to be more careful here.

Three-dimensional tracking resolution

Similarly, the authors now briefly compare their work to localisation microscopy, but not to the state-of-the-art there. Their conclusion is that they achieve comparable sensitivity to the papers they compare with. This is concerning: localisation microscopy is simple and easy to apply, and can give three dimensional information. No cavity is required, the approach can be applied in a conventional microscope, and it does not suffer from the dynamic range issues problematic here. The conclusion then is that, with all the addition effort and challenges, the current paper does not achieve better performance than easier to implement, widely used, techniques that suffer much fewer drawbacks. This illustrates why, in my view, the current work is not quite there for Nature Communications. Were it to show a substantial improvement in sensitivity over the state of the art, then things would get interesting.

I would reiterate from my previous review that optical tweezers based particle tracking can achieve far higher sensitivity than reported here (and does not require the averaging over seconds stated by the authors in their response). For instance [Chavez, Rev. Sci. Inst. 79 105104 (2008)] has around 10 fm/rtHz sensitivity, which corresponds to 0.6 picometers over the authors 0.3 ms measurement time. Clearly this is far superior to what the authors report. Of course, this is

achieved for much larger particles. Still, if the authors wish to make comparisons with the state-of-the-art (which I think is really needed for a paper like this), I would encourage them to be comprehensive, rather than picking only a limited set of particular comparisons.

Reply: Q/V sensitivity

We thank the referee for pointing out that we should compare the Q/V sensitivity of our Fabry-Pérot sensor more meticulously with the sensors' sensitivity of other groups.

As it is done for other sensor techniques, we could also improve the sensitivity of our sensor by exploiting the small mode volume of plasmonic nanoparticles. Therefore, to draw a reasonable comparison, we only compare our sensor with microspheres and microtoroids in an aqueous environment and without the combination with plasmonics.

The **microsphere sensors** in [2] have a Q factor of $2.6 \cdot 10^5$ to $6.4 \cdot 10^5$. For the specified microsphere geometries and used wavelengths, J. Dobrindt simulated the mode as well as the sensing volumes (see Table A.2. in reference [1]). The mode volume is defined as the volume of the mode located inside as well as outside of the microsphere. On the other hand, the sensing volume is only the evanescent part of the mode volume outside of the microsphere with a reduced electric field amplitude. The latter is responsible for the produced shift by the nanoparticle, namely the shift is proportional to the overlap of the nanoparticle with the sensing volume V_s . From the sensing and mode volume values in Table A.2. it can be concluded that the sensing volume is about 1.8 to 4.0 times larger than the mode volume. Thus, the sensitivity $Q/V_\lambda = Q \cdot \lambda^3/V_s$ of the used microspheres ranges from 26 to 95. In newer publications, the same research group could show that quality factors of up to $Q \approx 5 \cdot 10^6$ are possible[3, 4, 5]. Here, the sensing and mode volumes are not specified. By assuming similar sensing volumes as in Fig. ??, the maximal sensitivity of microspheres is at the order of $4.7 \cdot 10^2$. Here we can conclude that the microsphere sensitivity is about two orders of magnitude smaller than the sensitivity of our sensor ($Q \cdot \lambda^3/V_m = 4.7 \cdot 10^4$).

For the comparison with **microtoroid sensors**, we consider the results of different research groups (see Tabular 1). The best sensitivity reached with microtoroids is on the order of $Q \cdot \lambda^3/V_s \approx 9.7 \cdot 10^3 \dots 2.2 \cdot 10^4$. With our Fabry-Pérot sensor we don't have to distinguish between the sensing and the mode volume, since we have an open-access cavity with a mode formed between the two mirrors in open space such that the field maximum is directly accessible. Here, we reach a sensitivity of $Q \cdot \lambda^3/V_m \approx 4.7 \cdot 10^4$, which is about 2 to 5 times greater as the best sensitivity achieved with microtoroids.

In summary, the reviewer is right that our statement in the manuscript "The figure of merit for the sensitivity of this cavity, [...] is a factor ≈ 100 higher than those of optimized WGM resonators" (line 128-131) doesn't include the comparison with microtoroids. We want to apologize for this and changed the sentence to "The figure of merit for the sensitivity of this cavity, [...] is a factor ≈ 100 greater than those of optimized WGM microspheres and about 2 to 5 times greater than the best reported microtoroid sensor in aqueous solution."

We want to emphasize that we developed a new kind of sensor for nanoparticle sensing in an aqueous solution [6], which already for this first demonstration achieves better sensitivity than microspheres and microtoroids. In addition, we have not yet exhausted all possibilities in order to increase the sensitivity. For instance establishing lock-in techniques in order to reduce the measurement noise and exploiting nanoplasmonics in order to reduce the mode volume. Therefore, we see a great potential in our sensor for a further improvement of sensitivity and towards the detection of single molecules with a small molecular mass.

Publication	Quality factor Q	Mode volume V_m (μm^3)	Sensing volume V_s (μm^3)	Sensitivity $Q \cdot \lambda^3 / V_s$
Su et al. [7]	$5 \cdot 10^6$?	?	$\approx 2.5 \cdot 10^2?$
He at al. [8]	$6 \cdot 10^6$?	?	$\approx 3.1 \cdot 10^3?$
Swaim et al. [9]	$6.6 \cdot 10^5$	350	630...1400?	$\approx 2.2 \cdot 10^2 \dots 4.7 \cdot 10^2?$
Li et al. [10]	$1 \cdot 10^8$	800	1440...3200?	$\approx 9.7 \cdot 10^3 \dots 2.2 \cdot 10^4?$
Armani et al. [11]	$2 \cdot 10^8$?	?	$\approx 3.1 \cdot 10^3?$
Lu et al. [12]	$1 \cdot 10^8$?	?	$\approx 3.1 \cdot 10^3?$

Table 1: Sensitivity of different microtoroid sensors. For the estimation of the sensing volumes, a factor of 1.8 and 4.0 on the mode volume is assumed.

Reply: Three-dimensional tracking resolution

We thank the referee for pointing out that we should compare our sensor's localization and time resolution in a more extensive way with other localization techniques.

In general, besides all the known advantages, the **tracking of fluorophores** suffers from photobleaching and saturation. The former limits the localization and the latter the time resolution. Fig. 1 in reference [14] shows different microscopy techniques based on fluorophores and their resolution limits. The best spatial resolution of ≈ 10 nm can be achieved with iPALM, however the time resolution is on the order of 10 s. On the other hand, the (almost) best time resolution of ≈ 0.16 s have the sCMOS SMSN sensors and the localization resolution is ≈ 15 nm. Recently, a better spatial resolution of 2 nm was achieved with MINIFLUX [13] at a time resolution of $4 \cdot 10^{-4}$ s. However, the technique is limited to 2D, needs to use suitable fluorophores, and so far, only strongly restricted motion was observed and not free Brownian motion.

In tabular 2 other localization microscopy techniques are summarized.

Microscope	Particle radius (nm)	Time resolution (s)	Spatial resolution (nm)	Comment
Our sensor	60 (SiO ₂) current limit: 20	$3.3 \cdot 10^{-4}$	8...44	
MINFLUX[13]	ATTO 647N 850g/mol=0.85kDa	$4 \cdot 10^{-4}$ at photon count rate 0.4 MHz	2	Fluorescent beads. Label changes natural behavior. Limited to two dimensions.
TSUNAMI [15]	20...50	$5 \cdot 10^{-5}$ at photon count rate 1-2 MHz	16...35	Fluorescent beads. Label changes natural behavior.
iScat [16]	10...20 (Au)	$1 \cdot 10^{-3}$	1.9	Tracking in two dimensions. Strong scatterer is used.
iScat [17]	60 kDA ≈ 5 nm (PS)	$4 \cdot 10^{-2}$?	Tracking in two dimensions.
aiScat [18]	Ferritin proteins ≈ 2 nm (Au)	$2.5 \cdot 10^{-3}$	12	Tracking in two dimensions. Strong scatterer is used.
Fiber detector [19]	500 (PS)	$3 \cdot 10^{-8}$	0.1	Tracking in one dimension. objects are comparable or larger than the wavelength.
Plasmonic nanoaperture (indirect optical trapping) [20]	10 (PS)	$3.3 \cdot 10^{-2}$?	Tracking in two dimensions. Local increased temperature could change molecular/nanoparticle behaviour
Stereo darkfield interferometry [21]	200	0.1	0.14...0.31	Poor time resolution.
Camera-based [22]	3000	0.1	0.3	Very large magnetic particles and poor time resolution.
Photothermal tracking [23]	5 (Au)	$3.3 \cdot 10^{-2}$	20	Poor time resolution. Limited to good absorbing nanoparticles/molecules.

Table 2: Comparison between different tracking microscopes/sensors.

A comparison between the different methods and specific experiments is non-trivial, since they all have very different limitations and capabilities. When considering our main claim, namely the capability of fast three-dimensional tracking of unlabeled nanoparticles, we still achieve better performance than any previous work that we could find. Labeling, restrictions to one or two dimensions, or long averaging times represent fundamental limitations, which we avoid in our approach, and still achieve values comparable to the best ones reported to date. Here we restrict to small objects well within the Rayleigh regime. The example [19] mentioned by the referee involves micron-scale particles, where the scattering cross section is larger than the diffraction limit, such that very large signal levels are achieved. This experiment rather belongs to the field of optical positioning experiments of optically resolved objects, as it is done e.g. also in the field of cavity optomechanics and gravitational wave detection. Certainly, in this field, much higher position accuracy can be achieved (e.g. $\sim 10^{-18}$ m for micron-scaled oscillators). However, for the applications where our sensor can be of value - resolving dynamics and properties of nanomaterials and biomolecules - such large objects are of no use. We would like to emphasize the dramatic difficulty of measuring sub-wavelength nanoparticles, because the polarizability volume scales with particle size d as $1/d^3$ and Rayleigh scattering with $1/d^6$. The particles that we studied have a particle size of $d \sim \lambda/6$. Therefore, the polarizability volume is $\sim 200\times$ and the scattering cross section is $\sim 5 \times 10^4\times$ smaller than that of a wavelength-size particle. This obviously needs to be taken into account when comparing experiments.

In summary, we think it is valid to state that we achieve state-of-the-art performance, but now add explicitly the significant benefit of our work for label-free three-dimensional tracking: "This is on par with state-of-the art nanoparticle localization techniques on such short timescales [13, 15, 16, 18, 24, 25], but for the first time this is achieved for the localization in three dimensions and with label-free nanoparticles." (line 327-331)

Reviewer 2

The authors have addressed all of my comments in the revised version of the manuscript, and I am therefore very happy for the manuscript to be published in Nature Communications.

Reply: We thank the reviewer for recommending our work being published in Nature communications.

Reviewer 3

The authors have done a great job in addressing the comments and concerns of the Reviewers in the revised version of the manuscript. I think the techniques developed in this manuscript and the demonstrated performance of the system have the potential to find widespread use in label-free detection and characterization of particles as well as the label-free study of the dynamics of biologically relevant particles and processes. The paper is well-written, Supplement provides sufficient information to understand both experimental and numerical techniques used in the assessment of the performance of the device. In short, I am happy with the revised version of the manuscript and can recommend it for publication in Nature Communications.

Reply: We thank the reviewer for recommending our work being published in Nature communications.

Additional changes in the manuscript

We have noticed an error in the size distribution of sample B: In Supplementary Figure 10 the standard deviation of $\sigma_r = \sqrt{PDI} \cdot r_{\text{hydr}} = 3.8 \text{ nm}$ was wrongly calculated. With a PDI value of 0.016 and a z-average of 150.6 nm, the standard deviation is $\sigma_r = \sqrt{PDI} \cdot r_{\text{hydr}} = 9.5 \text{ nm}$. We apologize this mistake and as a result, we changed the following sections in the manuscript as well as in the supplementary information:

- Supplementary Figure 8: The determination of the effective refractive index for sample B. New result: $n_{\text{eff}} = 1.43 \pm 0.02$ (before: 1.46 ± 0.01).
- Manuscript Fig. 2d: The simulated curves and limits have been adjusted to the new effective refractive index.
- Manuscript line 260: The refractive index of the hydration shell has changed to 1.39 ± 0.04 (before: 1.46 ± 0.01).

A small mistake has also occurred in the calculation of the hydration shell refractive index error of sample A. Here the error of the nanoparticle's intrinsic radius was assumed twice too high (we have confused diameter and radius in the calculation). Therefore, we changed in line 260 $n_h = 1.40 \pm 0.04$ to half the error $n_h = 1.40 \pm 0.02$.

All the new changes in the manuscript are marked in blue and the changes of the first revision are still marked in red.

References

- [1] Jens M Dobrindt. Bio-sensing using toroidal microresonators Theoretical cavity optomechanics. 2012.
- [2] F. Vollmer, S. Arnold, and D. Keng. Single virus detection from the reactive shift of a whispering-gallery mode. *Proceedings of the National Academy of Sciences of the United States of America*, 105(52):20701–20704, 2008.
- [3] S. Arnold, D. Keng, S. I. Shopova, S. Holler, W. Zurawsky, and F. Vollmer. Whispering gallery mode carousel – a photonic mechanism for enhanced nanoparticle detection in biosensing. *Optics Express*, 17(8):6230, 2009.
- [4] Martin D. Baaske, Matthew R. Foreman, and Frank Vollmer. Single-molecule nucleic acid interactions monitored on a label-free microcavity biosensor platform. *Nature Nanotechnology*, 9(11):933–939, 2014.
- [5] Martin D. Baaske and Frank Vollmer. Optical observation of single atomic ions interacting with plasmonic nanorods in aqueous solution. *Nature Photonics*, 10(11):733–739, 2016.

- [6] Giuliano Zanchetta, Roberta Lanfranco, Fabio Giavazzi, Tommaso Bellini, and Marco Buscaglia. Emerging applications of label-free optical biosensors. *Nanophotonics*, 6(4):627–645, 2017.
- [7] Judith Su, Alexander F.G. Goldberg, and Brian M. Stoltz. Label-free detection of single nanoparticles and biological molecules using microtoroid optical resonators. *Light: Science and Applications*, 5(August 2015):2–7, 2016.
- [8] Lina He, Şahin Kaya Özdemir, Jiengang Zhu, Woosung Kim, and Lan Yang. Detecting single viruses and nanoparticles using whispering gallery microlasers. *Nature Nanotechnology*, 6(7):428–432, 2011.
- [9] Jon D. Swaim, Joachim Knittel, and Warwick P. Bowen. Detection of nanoparticles with a frequency locked whispering gallery mode microresonator. *Applied Physics Letters*, 102(18), 2013.
- [10] Bei Bei Li, William R. Clements, Xiao Chong Yu, Kebin Shi, Qihuang Gong, Yun Feng Xiao, and Oskar J. Painter. Single nanoparticle detection using split-mode microcavity Raman lasers. *Proceedings of the National Academy of Sciences of the United States of America*, 111(41):14657–14662, 2014.
- [11] Andrea M. Armani, Rajan P. Kulkarni, Scott E. Fraser, Richard C. Flagan, and Kerry J. Vahala. Label-free, single-molecule detection with optical microcavities. *Science*, 317(5839):783–787, 2007.
- [12] Tao Lu, Hansuek Lee, Tong Chen, Steven Herchak, Ji Hun Kim, Scott E. Fraser, Richard C. Flagan, and Kerry Vahala. High sensitivity nanoparticle detection using optical microcavities. *Proceedings of the National Academy of Sciences of the United States of America*, 108(15):5976–5979, 2011.
- [13] Yvan Eilers, Haisen Ta, Klaus C. Gwosch, Francisco Balzarotti, and Stefan W. Hell. MINFLUX monitors rapid molecular jumps with superior spatiotemporal resolution. *Proceedings of the National Academy of Sciences of the United States of America*, 115(24):6117–6122, 2018.
- [14] Kevin Welsher and Haw Yang. Imaging the behavior of molecules in biological systems: Breaking the 3D speed barrier with 3D multi-resolution microscopy. *Faraday Discussions*, 184:359–379, 2015.
- [15] Evan P. Perillo, Yen Liang Liu, Khang Huynh, Cong Liu, Chao Kai Chou, Mien Chie Hung, Hsin Chih Yeh, and Andrew K. Dunn. Deep and high-resolution three-dimensional tracking of single particles using nonlinear and multiplexed illumination. *Nature Communications*, 6(91), 2015.
- [16] Chia Lung Hsieh, Susann Spindler, Jens Ehrig, and Vahid Sandoghdar. Tracking single particles on supported lipid membranes: Multimobility diffusion and nanoscopic confinement. *Journal of Physical Chemistry B*, 118(6):1545–1554, 2014.
- [17] J. Ortega Arroyo, J. Andrecka, K. M. Spillane, N. Billington, Y. Takagi, J. R. Sellers, and P. Kukura. Label-free, all-optical detection, imaging, and tracking of a single protein. *Nano Letters*, 14(4):2065–2070, 2014.

- [18] Matz Liebel, James T. Hugall, and Niek F. Van Hulst. Ultrasensitive Label-Free Nanosensing and High-Speed Tracking of Single Proteins. *Nano Letters*, 17(2):1277–1281, 2017.
- [19] Isaac Chavez, Rongxin Huang, Kevin Henderson, Ernst Ludwig Florin, and Mark G. Raizen. Development of a fast position-sensitive laser beam detector. *Review of Scientific Instruments*, 79(10):2006–2009, 2008.
- [20] Zhe Xu, Wuzhou Song, and Kenneth B. Crozier. Direct Particle Tracking Observation and Brownian Dynamics Simulations of a Single Nanoparticle Optically Trapped by a Plasmonic Nanoaperture. *ACS Photonics*, 5(7):2850–2859, 2018.
- [21] Martin Rieu, Thibault Vieille, Gaël Radou, Raphaël Jeanneret, Nadia Ruiz-Gutierrez, Bertrand Ducos, Jean-François Allemand, and Vincent Croquette. Parallel, linear, and sub-nanometric 3D tracking of microparticles with Stereo Darkfield Interferometry. *Science Advances*, 7(6):eabe3902, 2021.
- [22] Alexander Huhle, Daniel Klaue, Hergen Brutzer, Peter Daldrop, Sihwa Joo, Oliver Otto, Ulrich F. Keyser, and Ralf Seidel. Camera-based three-dimensional real-time particle tracking at kHz rates and Ångström accuracy. *Nature Communications*, 6, 2015.
- [23] David Lasne, Gerhard A. Blab, Stéphane Berciaud, Martin Heine, Laurent Groc, Daniel Choquet, Laurent Cognet, and Brahim Lounis. Single Nanoparticle Photothermal Tracking (SNaPT) of 5-nm gold beads in live cells. *Biophysical Journal*, 91(12):4598–4604, 2006.
- [24] Thomas T. Perkins. Ångström-Precision Optical Traps and Applications*. *Annual Review of Biophysics*, 43(1):279–302, 2014.
- [25] Carlo Manzo and Maria F. Garcia-Parajo. A review of progress in single particle tracking: From methods to biophysical insights. *Reports on Progress in Physics*, 78(12), 2015.

REVIEWERS' COMMENTS

Reviewer #1:

The authors have done some further literature surveys and made some minor revisions following my feedback.

In my mind, the conclusions of the literature survey and revisions are to reinforce my view as accurate that the paper probably does not warrant publication in Nature Communications. Indeed, the minor revisions are to greatly revise down the significance of the work (e.g. from that there work is 100 times superior to all previous cavity based sensors, to being a factor of a few better than a narrowly defined set of sensors).

I feel that the authors literature survey remains relatively narrow. E.g. why exclude experimentally demonstrated optoplasmonic cavities from the comparison? The justification to exclude them because the authors could later include plasmonics to their cavities appears to me very weak. What about photonic crystal cavities? Or hybrid plasmonic-photonic crystal cavities? Micromachines 2020, 11(1), 72; <https://doi.org/10.3390/mi11010072> is a good recent review of photonic crystal cavities for sensing. Table 1 of this review has several structures with figures of merit ($Q \lambda^3/V$) orders of magnitude higher than the authors demonstrate. It seems like the authors have taken specific examples of cavities that suit their arguments, rather than doing a genuine thorough comparison.

I stand by my assessment that other 3D tracking techniques already in use are far more practical and easily implementable than the authors solution. These other solutions further do not suffer from the severe dynamic range issue for the z axis in the authors technique. From my perspective, this dynamic range issue essentially precludes any practical particle tracking in that dimension, which I think clearly shows through in Fig. 3d and e of the manuscript.

I would emphasize again that I do believe that the authors work is a nice step towards interesting applications. I just don't think they've got there yet. It is remarkable that, using a high quality extremely small mode volume cavity they are only able to achieve comparable performance to techniques that avoid these complications altogether (like the MINFLUX technique they now reference in their rebuttal). If they were to achieve performance significantly beyond these techniques, and included a convincing and comprehensive analysis evidencing this, I would be an avid supporter of the work.

Reviewer #3:

Some minor things that I believe the authors should address in the final form of the manuscript before decision on publication:

1. I realized that the authors mention the WGM Raman laser-based sensing of nanoparticles but fail to cite the pioneering works that were published before the one they cite. I believe these milestone papers should be cited. These papers can be cited together with Ref. 30: i) PNAS 111, E3836-E3844 (2014), and ii) Nat. Nanotechnol. 6, 428-432 (2011). These works are both on WGM microlaser based nanoparticle detection and characterization.
2. References 19 and 31 are the same. Zhu et. al.
3. Reference 29: Although this paper has been published in Science, the results have not been independently confirmed. There are questions about the validity of the results and the processes which lead to enhanced sensitivity in that paper. There should be an erratum in Science about this work and a critics published in Optics Express. The authors should reconsider citing this paper - if they decide to keep this work in the reference list, I suggest that they also cite the corrections and the paper in Optics Express: Optics Express Vol. 18, pp. 281-287 (2010)
4. Please provide the estimated polarizability values of the detected nanoparticles.

Response to the reviewers

Reviewer 1

The authors have done some further literature surveys and made some minor revisions following my feedback.

Reviewer Point P 1.1 — In my mind, the conclusions of the literature survey and revisions are to reinforce my view as accurate that the paper probably does not warrant publication in Nature Communications. Indeed, the minor revisions are to greatly revise down the significance of the work (e.g. from that there work is 100 times superior to all previous cavity based sensors, to being a factor of a few better than a narrowly defined set of sensors).

Reply: We would like to emphasize that our comparison was made for the context of sensors that are in principle capable of 3D tracking as well as characterization of intrinsic properties of nanosystems, a crucial capability which to our knowledge has not been shown to date in combination, but which might be possible to do with the WGM sensors to which we mostly refer to.

Reviewer Point P 1.2 — I feel that the authors literature survey remains relatively narrow. E.g. why exclude experimentally demonstrated optoplasmonic cavities from the comparison? The justification to exclude them because the authors could later include plasmonics to their cavities appears to me very weak. What about photonic crystal cavities? Or hybrid plasmonic-photonic crystal cavities? *Micromachines* 2020, 11(1), 72; <https://doi.org/10.3390/mi11010072> is a good recent review of photonic crystal cavities for sensing. Table 1 of this review has several structures with figures of merit ($Q \lambda^3/V$) orders of magnitude higher than the authors demonstrate. It seems like the authors have taken specific examples of cavities that suit their arguments, rather than doing a genuine thorough comparison.

Reply: As mentioned above, we believe that it makes sense to compare with sensors that can possibly perform similar sensing measurements as demonstrated in our work, 3D tracking, and quantitative characterization of intrinsic nanosystem properties such as polarizability, size, hydrodynamic radius, etc. Nanobeam cavities are typically used for refractive index sensing and detection of nanosystems, and achieve indeed higher figures of merit therefore. But they are neither capable of 3D tracking, nor of quantitative single NP characterization, given the extremely small active sensing volume and proximity of surfaces. The same is true for nanoplasmonic sensors.

Reviewer Point P 1.3 — I stand by my assessment that other 3D tracking techniques already in use are far more practical and easily implementable than the authors solution. These other solutions further do not suffer from the severe dynamic range issue for the z axis in the authors technique. From my perspective, this dynamic range issue essentially precludes any practical particle tracking in that dimension, which I think clearly shows through in Fig. 3d and e of the manuscript.

Reply: Indeed, in our current work, we are limited by a very small dynamic range for the z-axis. But as we point out in the discussion section, performing a two-frequency measurement, which leads to

a spatial beating pattern, the z-range can be extended to several micrometers, similar to the other dimensions, such that the tracking volume can be substantially extended in future work using the same technique.

Reviewer Point P 1.4 — I would emphasize again that I do believe that the authors work is a nice step towards interesting applications. I just don't think they've got there yet. It is remarkable that, using a high quality extremely small mode volume cavity they are only able to achieve comparable performance to techniques that avoid these complications altogether (like the MINIFLUX technique they now reference in their rebuttal). If they were to achieve performance significantly beyond these techniques, and included a convincing and comprehensive analysis evidencing this, I would be an avid supporter of the work.

Reply: We would like to emphasize that the main result of our work is that 3D tracking can be combined with and contributes to the quantitative characterization of intrinsic properties of an unmodified sample. This is e.g. not possible with fluorescence labeling (e.g. MINIFLUX) or other non-quantitative methods. We are convinced that this is an important capability for investigating bio-molecular nanosystems in the future.

Reviewer 2

-

Reply: -

Reviewer 3

Reviewer Point P 3.1 — Some minor things that I believe the authors should address in the final form of the manuscript before decision on publication:

Reply: We thank the referee for the following valuable and supporting comments.

Reviewer Point P 3.2 — I realized that the authors mention the WGM Raman laser-based sensing of nanoparticles but fail to cite the pioneering works that were published before the one they cite. I believe these milestone papers should be cited. These papers can be cited together with Ref. 30: i) PNAS 111, E3836-E3844 (2014), and ii) Nat. Nanotechnol. 6, 428-432 (2011). These works are both on WGM microlaser based nanoparticle detection and characterization.

Reply: We thank the Reviewer for this advice and have added the two references as suggested.

Reviewer Point P 3.3 — References 19 and 31 are the same. Zhu et. al.

Reply: We thank the reviewer for this comment and have removed the double reference.

Reviewer Point P 3.4 — Reference 29: Although this paper has been published in Science, the results have not been independently confirmed. There are questions about the validity of the results and the processes which lead to enhanced sensitivity in that paper. There should be an erratum in Science about this work and a criticism published in Optics Express. The authors should reconsider citing this paper— if they decide to keep this work in the reference list, I suggest that they also cite the corrections and the paper in Optics Express: Optics Express Vol. 18, pp. 281-287 (2010)

Reply: We thank the referee for this comment. We agree that it is not appropriate to cite reference 29 and therefore have removed this reference.

Reviewer Point P 3.5 — Please provide the estimated polarizability values of the detected nanoparticles.

Reply: We have now given the values for the polarizability of sample A and sample B in the main text as wished (text color: blue).

Additional changes in the manuscript

- We noticed an inaccuracy in our comparison of the figure of merit of microtoroid cavities with our device, where we inadvertently compared the cooperativity of our sensor in air with the cooperativity of the best reported microtoroids in water. We corrected this mistake and now write "on par with the best reported microtoroid sensors" (text color: blue).
- During the final revision we discovered the following paper, which investigates 3D tracking of nanoparticles in a hollow core fiber, and which we now cite in the manuscript: Jiang et al., Nanoscale, 12(5):3146-3156,2020 (text color: blue, reference 7). Concerning the comparison with three-dimensional tracking it is helpful to cite this paper. However, with this sensor it is not possible to combine three-dimensional tracking with quantitative characterization measurements and, in addition, the sensitivity is lower compared to our sensor.